# Altered Expression of *TMEM43* Causes Abnormal Cardiac Structure and Function in Zebrafish

**DOI:** 10.3390/ijms23179530

**Published:** 2022-08-23

**Authors:** Miriam Zink, Anne Seewald, Mareike Rohrbach, Andreas Brodehl, Daniel Liedtke, Tatjana Williams, Sarah J. Childs, Brenda Gerull

**Affiliations:** 1Comprehensive Heart Failure Center, Department of Internal Medicine I, University Hospital Würzburg, 97080 Würzburg, Germany; 2Erich and Hanna Klessmann Institute, Heart and Diabetes Center NRW, University Hospital of the Ruhr-University Bochum, 32545 Bad Oeynhausen, Germany; 3Institute for Human Genetics, Biocenter, Julius-Maximilians-University Würzburg, 97074 Würzburg, Germany; 4Department of Biochemistry and Molecular Biology, Alberta Children’s Hospital Research Institute, University of Calgary, Calgary, AB T2N 4N1, Canada

**Keywords:** TMEM43, arrhythmogenic cardiomyopathy, zebrafish, CRISPR/Cas9

## Abstract

Arrhythmogenic cardiomyopathy (ACM) is an inherited heart muscle disease caused by heterozygous missense mutations within the gene encoding for the nuclear envelope protein transmembrane protein 43 (*TMEM43*). The disease is characterized by myocyte loss and fibro-fatty replacement, leading to life-threatening ventricular arrhythmias and sudden cardiac death. However, the role of TMEM43 in the pathogenesis of ACM remains poorly understood. In this study, we generated cardiomyocyte-restricted transgenic zebrafish lines that overexpress *eGFP*-linked full-length human wild-type (WT) *TMEM43* and two genetic variants (c.1073C>T, p.S358L; c.332C>T, p.P111L) using the Tol2-system. Overexpression of WT and p.P111L-mutant TMEM43 was associated with transcriptional activation of the mTOR pathway and ribosome biogenesis, and resulted in enlarged hearts with cardiomyocyte hypertrophy. Intriguingly, mutant p.S358L TMEM43 was found to be unstable and partially redistributed into the cytoplasm in embryonic and adult hearts. Moreover, both TMEM43 variants displayed cardiac morphological defects at juvenile stages and ultrastructural changes within the myocardium, accompanied by dysregulated gene expression profiles in adulthood. Finally, CRISPR/Cas9 mutants demonstrated an age-dependent cardiac phenotype characterized by heart enlargement in adulthood. In conclusion, our findings suggest ultrastructural remodeling and transcriptomic alterations underlying the development of structural and functional cardiac defects in TMEM43-associated cardiomyopathy.

## 1. Introduction

Arrhythmogenic cardiomyopathy (ACM) is a hereditary heart muscle disease leading to life-threatening ventricular arrhythmias and sudden cardiac death, often in young people between adolescence and mid-adulthood. Pathologically, ACM is characterized by progressive loss of myocytes and fibro-fatty replacement of biventricular myocardium [1,2,3]. Despite uncovering disease-linked variants in more than 25 different genes, a significant number of patients do not carry pathogenic variants, and, conversely, many ACM patients bearing variants with high penetrance do not present cardiac symptoms, even at older ages. This indicates a complex and heterogeneous etiology, with polygenic, epigenetic, and environmental factors underlying the development of ACM [4,5]. As a consequence, clinical diagnosis of ACM remains challenging, even with the application of the task force criteria or Padua criteria. The diagnosis depends on a broad clinical assessment of patients with a combination of various types of information, including electro- and echocardiographic, histopathological, and genetic data. Moreover, it is important to consider differential diagnosis that encompasses manifestations in an athlete’s heart, cardiac syndromes orare common with other cardiomyopathies [1,4,6]. The majority of ACM-linked mutations are located in gene-encoding proteins of desmosomal and other intercellular junctional complexes, such as plakophilin-2 [7] or desmoplakin [8]. However, mutations residing in non-desmosomal gene-encoding proteins with biological functions in nucleoskeletal [9,10,11] and cytoskeletal architecture [12,13], calcium handling [14,15], sodium transport [16], and cytokine signaling [17] have also been described. A heterozygous variant (c.1073C>T, p.S358L) within the transmembrane protein 43 (*TMEM43*) gene has been unequivocally identified to cause a fully penetrant and sex-influenced form of ACM in unrelated patients around the globe [18,19,20,21,22]. Other rare TMEM43 variants (NM_024334.3(TMEM43): c.265G>A, p.V89M; c.896G>C, p.R299T; c.331C>G, p.P111L) have also been proposed to be causative for ACM [23,24]. The heterozygous TMEM43 missense variants c.253G>A (p.E85K) and c.271A>G (p.I91V) have been reported to cause Emery–Dreifuss muscular dystrophy type 7 [25,26]. Recently, a nonsense TMEM43 variant (c.1114C>T, p.A372X) has been identified in two Asian families with inherited late-onset auditory neuropathy spectrum disorder (ANSD) without causing any cardiac abnormalities [27]. TMEM43 is an integral protein of the inner nuclear membrane, but is also found in adherens and composite junctions of cardiomyocytes (CMs) and epithelial cells [28,29]. Structurally, the protein contains four transmembrane domains (TMD) and a large hydrophilic domain that is exposed to the endoplasmic reticulum (ER). TMEM43 has been shown to interact with components of the linker of nucleoskeleton and cytoskeleton (LINC) complex, such as emerin and lamins, and thus is involved in the organization of the nuclear membrane [28]. To date, several model systems, including induced pluripotent stem cell derived CMs (iPSC-CMs), transgenic mice and rats carrying the *TMEM43* c.1073C>T, p.S358L variant, or *Tmem43* knock-out mutations have been generated to investigate the molecular mechanism underlying ACM [22,30,31,32,33,34,35]. However, the associated phenotypes among the animal models are highly variable, suggesting a controversial role of Tmem43 in murine heart. Although previous studies imply that the canonical WNT, NF-κB, and TGF-β signaling pathways may play a role in the pathogenesis of TMEM43-associated ACM, the exact molecular mechanisms have not yet been fully elucidated [30,33]. Here, we report, for the first time, the generation and characterization of transgenic zebrafish lines carrying either different ACM-associated TMEM43 variants (c.332C>T, p.P111L; c.1073C>T, p.S358L) or *tmem43* knock-out mutations. Our data demonstrate that cardiac overexpression of human *TMEM43* may affect cardiac development in zebrafish at an early developmental stage. Expression of either wild-type (WT) or TMEM43 p.P111L provoke hypertrophic ventricular CMs and, subsequently, result in enlarged hearts through hyperactivation of the mTOR pathway and ribosome biogenesis. Moreover, we demonstrate that TMEM43 variants lead to ultrastructural remodeling and transcriptomic changes in ventricular tissue. Loss of function *tmem43* mutants have been established by using CRISPR/Cas9 and display no impact on embryonic cardiac development or contractility, but leads to late-onset enlargement of the heart. Our data strongly indicate that gain and loss of function of *TMEM43*/*tmem43* alter cardiac morphology and function in zebrafish, thereby resembling pathophysiology in ACM patients.

## 2. Results

### 2.1. Conservation and Expression of TMEM43 Ortholog in Zebrafish

To assess the suitability of the zebrafish as a model system for TMEM43-associated cardiomyopathy, we determined the homology of zebrafish tmem43 with the corresponding sequences of humans and various other species. The zebrafish possesses a single tmem43 ortholog (ZDB-GENE-040426-2348) encoding a 391 amino acid protein (UniProt: A8KBQ5). BLAST analysis of human TMEM43 (RefSeq: NP_077310.1; UniProt: Q9BTV4) amino acid sequence against Danio rerio (GenBank accession: AAI54205.1), Xenopus tropicalis (XP_002938277), Gallus gallus (XP_414378), Canis lupus familiaris (XP_541751), Pan troglodytes (XP_001157301), Mus musculus (NP_083042.1), and Rattus norvegicus (NP_001007746) determined evolutionary conservation throughout vertebrate evolution (Appendix A). Protein sequence alignment using the Needleman–Wunsch algorithm resulted in 50.2% amino acid identification between human and zebrafish TMEM43, indicating a high level of conservation. The serine residue at amino acid position 358 is conserved in all species, whereas the proline residue at position 111 showed weak conservation and is replaced by a histidine residue in zebrafish. Since tmem43 has not been functionally investigated in zebrafish, we first determined the expression profile of tmem43 during embryonic stages and in adult zebrafish tissues by performing semi-quantitative reverse-transcription PCR (RT-PCR) using tmem43-specific primers and cDNA obtained from whole embryos or a set of adult organs (Appendix A). Expression of tmem43 can be detected in all organs considered including heart, liver, eye, ovary/testis, brain, and skeletal muscle at 24 hours post fertilization (hpf), 2 and 3 days post-fertilization (dpf), implying a rather broad and ubiquitous expression. The spatial expression pattern of tmem43 by whole-mount in situ hybridization (WISH) revealed a diffuse signal in the head region at 24 hpf (Appendix A). From 2 to 3 dpf, additional signals appeared in the heart, the pectoral fin, and the otic vesicle (Appendix A). The tmem43 sense probe (negative control) did not detect any target in the same developmental stages, confirming the specificity of the antisense probe (Appendix A).

### 2.2. In Silico Modeling of the p.S358L Variant Predicts Conformational Change of TMEM43

In addition to the previously described deleterious TMEM43 variant c.1073C>T (p.S358L) [18,19,21], another novel TMEM43 variant (c.332C>T, p.P111L) was reported in one affected individual with ACM [24]. This missense variant was considered to be relevant because it was absent in 300 healthy controls and was classified possibly damaging by the prediction program PolyPhen-2. Recently, two further ACM-associated variants of TMEM43 (VCV001423775.1: c.331C>G, p.P111A; VCV001516992.1: c.332C>A, p.P111Q) have been reported at position 111 (ClinVar-database; April/2022). p.P111A and p.P111Q variants are absent in the population database (gnomAD v2.1.1) and very rare for the p.P111L variant (Appendix A). However, different prediction programs for the effect of missense changes on protein structure and function suggest that all three variants at p.P111 are likely to be tolerated, whereas ClinPred [36] classifies p.P111A as likely pathogenic (Appendix A). To analyze the structural effect of the different amino acid substitutions, we modeled WT, p.S358L, p.P111A, p.P111Q, and p.P111L TMEM43 proteins in silico (Figure 1). The p.S358L variant in TMD3 potentially alters TMEM43 conformation and resulted in a lower structural compactness compared to WT TMEM43 (Figure 1A,B). In contrast, the p.P111 variants appeared to have minor effects on TMEM43 protein structure, as there are little physiochemical differences between the proline residue and the substituents (Figure 1C–F).

### 2.3. The p.S358L Variant Alters Cellular Localization of TMEM43 in Zebrafish Cardiomyocytes

To study the role of TMEM43 in the zebrafish heart, we established the first transgenic zebrafish model with cardiac-restricted overexpression of eGFP-linked full-length human WT TMEM43 (TMEM43-WT) and the respective genetic variants p.S358L (TMEM43-S358L) and p.P111L (TMEM43-P111L) using the Tol2 transposon system (Appendix A) [37,38]. All analyses were performed on progeny of F2 incrosses. First, we examined the expression levels of the transgene in adult ventricles by quantitative RT-PCR (qPCR) with probes detecting exclusively the human TMEM43 gene (Figure 2A). Both TMEM43 variant lines demonstrated a significant reduction in transgene mRNA expression levels compared to the TMEM43-WT (TMEM43-P111L: 58.1% and TMEM43-S358L: 16.0% of WT, respectively). Expression level of the endogenous tmem43 gene was slightly reduced in all transgenic lines compared to the non-transgenic control (NTG; Appendix A). Subsequently, Western blot analysis revealed a 2.63-fold lower expression of the fusion-protein in TMEM43-S358L ventricles compared to WT, whereas the TMEM43-P111L exhibited a similar expression level to the WT of the fusion-protein (Figure 2B,C). Immunofluorescent staining of transgenic embryos and heart tissue sections of 5-month-old fish revealed TMEM43 signals at the nuclear membrane and the endoplasmic reticulum (ER) in TMEM43-WT and TMEM43-P111L ventricles, whereas TMEM43-S358L displayed a partial delocalization of TMEM43 with a diffuse distribution throughout the cytosol and an overall weaker expression at the nuclear membrane (Figure 2D,E). Additionally, no signal in the ER was detected in p.S358L mutant ventricles, indicating an unstable p.S358L mutant protein with redistribution to the cytoplasm.

### 2.4. Mutant TMEM43 Impairs Cardiac Development and Alters Cardiac Function at 3 dpf

To determine if WT and mutant TMEM43 overexpression alters cardiac development, we evaluated cardiac morphology and function in transgenic zebrafish embryos at 3 dpf using microscopy techniques. Phenotyping unveiled an increased proportion of cardiac developmental defects in TMEM43 variant overexpressing embryos compared to TMEM43-WT and NTG control (Figure 3A). Moreover, the number of embryos displaying two or more defects is increased. The predominant defects are incorrect looping and pericardial edema (Figure 3B). Next, we looked at the cardiac chamber structure of embryos without any obvious developmental defects, and found these appeared normal on hematoxylin and eosin (H&E) stained sagittal sections of embryos at 3 dpf, with clearly distinguishable atrium, ventricle, and Bulbus arteriosus (Figure 3C). The inner surface of the ventricle developed trabeculation and the outer ventricular wall comprised one- to two-cell layers. Furthermore, we assessed ventricular contractility by measuring fractional shortening (FS) of morphologically normal p.S358L and p.P111L mutant embryos compared to TMEM43-WT and NTG controls. Increased FS values were found in TMEM43-S358L, but not in TMEM43-P111L and TMEM43-WT compared to NTG control (TMEM43-S358L: 20.84 ± 0.54%, *p* = 0.0319; TMEM43-WT: 17.30 ± 0.39%, *p* = 0.0684; TMEM43-P111L: 17.90 ± 0.47%, *p* < 0.9999; NTG: 18.77 ± 0.46%; Figure 3D). Together these data suggest that overexpression of mutant TMEM43 impairs cardiac development and alters function in p.S358L mutant zebrafish embryos.

### 2.5. Cardiac-Restricted Overexpression of TMEM43-WT and TMEM43-P111L Results in Hypertrophied Embryonic and Adult Hearts

Immunofluorescence-staining was performed to determine ventricular CM size and proliferation capacity in TMEM43 overexpressing lines. Staining of myocardial cell borders using an Alcam antibody (Figure 4A) revealed significantly increased CM areas in TMEM43-WT and the TMEM43-P111L lines compared to the NTG control (TMEM43-WT: 69.78 ± 2.69 µm^2^, *p* = 0.0128; TMEM43-P111L: 88.05 ± 2.56 µm^2^, *p* < 0.0001; NTG: 57.51 ± 2.51 µm^2^; Figure 4B). However, when comparing the TMEM43-S358L transgenic line to the NTG control this effect was not observed (TMEM43-S358L: 67.91 ± 2.66 µm^2^, *p* = 0.0915). Furthermore, we investigated whether CM hypertrophy in TMEM43-WT and TMEM43-P111L is associated with organ hypertrophy, which could be confirmed by significantly larger ventricular sizes measured by the end-diastolic ventricular area (TMEM43-WT: 12,307 ± 101.7 µm^2^, *p* < 0.0001; TMEM43-S358L: 10,977 ± 108.2 µm^2^, *p* = 0.3587; TMEM43-P111L: 11,809 ± 147.8 µm^2^, *p* < 0.0001; NTG: 10,536 ± 262.1 µm^2^; Figure 4C). Next, we also examined cell proliferation using the DNA replication marker PCNA (proliferating cell nuclear antigen) to detect ventricular cells in G1- and S-phase at 3 dpf (Figure 4D). Overall, proliferation was not affected in TMEM43-WT and TMEM43-P111L embryos compared to NTG, whereas the TMEM43-S358L line displayed a slightly increased cell proliferation rate (TMEM43-WT: 18.47 ± 3.31%, *p* = 0.3399; TMEM43-P111L: 10.58 ± 2.26%, *p* > 0.9999; TMEM43-S358L: 21.72 ± 2.96%, *p* = 0.0303; NTG: 10.36 ± 2.03%). Finally, quantification of cells in mitosis using phosphorylated Histone H3 (pHH3) on isolated embryonic hearts at 3 to 5 dpf showed no changes of cell proliferation capacity between all genotypes over time (Figure 4E).

To further determine the effect of TMEM43-WT and TMEM43 variants on adult hearts, we examined heart morphology in >5-month-old fish. Male and female TMEM43 transgenic zebrafish survived until adulthood. Quantification of the ventricular area of dissected hearts at >5 months of age showed that the ventricles in TMEM43-WT and TMEM43-P111L transgenic individuals remain enlarged compared to the NTG control. In particular, TMEM43-WT hearts showed a significantly increased ventricular area to body length ratio when compared to NTG hearts (TMEM43-WT: 0.0154 ± 0.0005 mm, *p* = 0.0027; TMEM43-P111L: 0.0151 ± 0.0007 mm, *p* = 0.0612; TMEM43-S358L: 0.0119 ± 0.0004 mm, *p* = 0.5081; NTG: 0.0131 ± 0.0004 mm; Figure 5A(a–d),B). There were no alterations in body length between the different genotypes or genders (Figure 5C). Heart tissue histology stained with H&E revealed no overt abnormalities (Figure 5A(e–h)). Moreover, there was no evidence of myocardial fibrosis based on Masson’s trichrome staining (MTC; Figure 5A(i–l)). Taken together, these data indicate that CM hypertrophy and subsequently enlarged hearts in these lines may be caused by overexpression of intact TMEM43 and is likely not related to genetic alterations. As ACM is often associated with arrhythmias, we conducted electrocardiogram (ECG) analysis to assess cardiac electrophysiology. Minor irregularities were observed in all genotypes, probably due to anesthesia, but no significant cardiac arrhythmias could be measured (Appendix A). Furthermore, the mean duration of RR intervals (Appendix A), as well as the average heart rate was similar in all lines (Appendix A). These data suggest that neither the altered cardiac morphology, nor the genetic alterations in TMEM43 appear to affect the conduction system.

### 2.6. Transcriptomic Analysis Reveals Regulation of Various Pathways including Cell Growth and Hypertrophy

We determined enlarged hearts in TMEM43-WT and TMEM43-P111L, that persist until adulthood. Thus, to examine mechanisms resulting in increased cell growth, we analyzed gene expression patterns of 5-month-old zebrafish by performing RNA-seq analyses.

Principal component analysis (PCA) showed four distinct clusters of samples, suggesting a significant differentiation between groups (Appendix A). We next performed analysis of differentially expressed genes (DEG) and compared the gene expression profiles of NTG control hearts with those of TMEM43-WT (2301 dysregulated genes; Appendix A), TMEM43-P111L (2078 dysregulated genes; Appendix A) and TMEM43-S358L (1863 dysregulated genes; Appendix A). When comparing “NTG vs. TMEM43-WT” with “NTG vs. TMEM43-P111L”, 1099 common significantly dysregulated genes were found. Analysis of enriched Kyoto Encyclopedia of Genes and Genomes (KEGG) pathways using ShinyGO v0.76 (false discovery rate (FDR) < 0.1) revealed the association of DEGs in aminoacyl-tRNA biosynthesis, ribosome biogenesis in eukaryotes, and mTOR signaling (Figure 4D), which all are known to play a pivotal role for cell growth [39]. Hierarchical clustering of the DEGs involved in these enriched KEGG pathways showed that these genes were mainly upregulated in TMEM43-WT and TMEM43-P111L transgenic lines (Figure 4E). To investigate mechanisms altered by the p.S358L variant, the 583 genes exclusively differently expressed in TMEM43-S358L compared to NTG ventricles were analyzed for enriched pathways. Interestingly, no significantly enriched KEGG pathways could be detected. Furthermore, the comparison of the 652 DEGs of “TMEM43-WT vs. TMEM43-S358L” also did not reveal any significantly enriched KEGG pathways. This indicates that when these subsets of DEGs were included, there were no obvious dysregulated pathways at the transcriptomic level in the TMEM43-S358L transgenic line.

### 2.7. Altered Ultrastructure of Ventricular Tissue in Adult TMEM43 Expressing Transgenic Zebrafish

To gain more insights at ultrastructural level, we performed transmission electron microscopy (TEM) on dissected ventricles of zebrafish at 5 months post-fertilization (mpf). We observed no significant differences between the genotypes in nuclear size and shape (Figure 6A–C). However, mitochondrial morphology was different, with reduced mitochondrial area in both TMEM43 variant transgenic lines compared to the TMEM43-WT and NTG (Figure 6D), and significantly altered mitochondrial circularity in TMEM43-WT and TMEM43-S358L ventricles compared to NTG (Figure 6E). Analysis of the 314 DEGs shared exclusively by “NTG vs. TMEM43-P111L” and “NTG vs. TMEM43-S358L”, identified 19 significantly enriched KEGG pathways (FDR < 0.1) pertaining to metabolic pathways (Figure 6F). Hierarchical clustering of the DEGs involved in these metabolic pathways revealed mainly downregulated genes in TMEM43-P111L and TMEM43-S358L (Figure 6G). Thus, we assume that the differences in mitochondrial morphology might provoke the observed alterations of metabolic genes in the TMEM43 variant lines. Longitudinal sections through sarcomeres revealed preserved sarcomeric organization in all zebrafish lines, with clearly defined M-lines and Z-discs and parallel aligned myofibrils (Figure 6H). Nonetheless, measurement of Z-line distance revealed reduced sarcomere length in all transgenic lines compared to the NTG control (Figure 6I). As TMEM43 has previously been reported to localize at the intercalated discs [29], we had a closer look at the desmosomes (Figure 6J). However, we could not detect any alterations of overall desmosome width by quantifying the distance between the plasma membranes (Figure 6K).

### 2.8. Generation of CRISPR/Cas9-Induced tmem43 Knock-Out Zebrafish Lines

We next attempted to investigate the role of Tmem43 for cardiac morphology using genetic loss of function via CRISPR/Cas9. We designed single-guide RNAs (sgRNAs) according to previously published guidelines [40] using ZiFit Targeter (http://zifit.partners.org/ZiFiT/ChoiceMenu.aspx (accessed on 27 July 2017)) [41,42]. A sequence at the beginning of exon 2 of tmem43 (genome assembly Zv9: ENSDARE00000787164) was targeted. The efficiency of the sgRNA was assessed, and F0 siblings were raised to adulthood and outcrossed to WT fish. To check successful introduction and germline transmission of indels, gDNA of the F1 progeny was sequenced. Three different alleles were recovered that are predicted to shift the reading frame and lead to a truncated protein, tmem43^wue4^, tmem43^wue5^, and tmem43^wue6^ (Appendix A). Endogenous tmem43 transcripts were reduced 70% in skeletal muscle tissue of homozygous mutants (tmem43^wue4/wue4^), indicating that most of the tmem43 mRNA is degraded by NMD (Appendix A).

### 2.9. Homozygous tmem43 Mutants Develop a Late-Onset Cardiac Phenotype

Homozygous tmem43 mutants from all lines were viable through adulthood and fertile. We then proceeded to analyze tmem43 mutant embryos at 3 dpf for cardiac parameters. As all three tmem43 CRISPR alleles are predicted to cause loss of function of Tmem43 we generated compound heterozygous knock-out animals. Embryos of tmem43^wue5^/tmem43^wue6^ were morphologically and functionally normal, indicated by normal cardiac phenotypes and unchanged FS values (Figure 7A–C). However, homozygous adult tme43^wue4/wue4^ zebrafish developed enlarged hearts, as indicated by significantly increased VA/BL (tmem43^wue4/wue4^: 0.01976 ± 0.00096 µm^2^, tmem43^wue4/+^: 0.01536 ± 0.00086 µm^2^, *p* = 0.0121; WT: 0.01605 ± 0.00099 µm^2^, *p* = 0.0442; Figure 7D(a–c) and Figure 7E). Although, H&E and MTC staining on myocardial sections revealed normal muscle structure (Figure 7D(d–i)). In addition, there were no changes in body length among the different genotypes (Figure 7F). In summary these data indicate that reduction in Tmem43 function in zebrafish mutants leads to late-onset cardiac enlargement.

## 3. Discussion

In human patients, mutations in the nuclear envelope protein TMEM43 are responsible for severe diseases, including ACM type 5, a devastating cardiomyopathy that causes malignant arrhythmias and heart failure [18,26,27]. Despite efforts in recent in vivo studies using transgenic, knock-in, or knock-out rodent models, the pathogenic mechanisms of TMEM43-associated ACM remain still poorly understood [22,30,31,32,33,34]. In this study, we generated the first transgenic zebrafish model expressing two different potential pathogenic variants found in human patients under a heart-specific promoter and generated genetic mutants of *tmem43* in zebrafish. We subsequently characterized the zebrafish lines from early embryonic stages to adulthood. Overexpression of TMEM43-WT and TMEM43-p.P111L in CMs promotes hypertrophic cardiac growth without affecting cellular structure and function. In contrast, the TMEM43-S358L transgenic line shows altered expression and localization of TMEM43 in embryonic and adult hearts suggesting an unstable and delocalized protein. This was further supported by in silico analysis of TMEM43-p.S358L protein structure, indicating structural instability and a deleterious effect [30]. Moreover, we found that TMEM43 variants could impair cardiac development and lead to differences in myocardial gene expression. Interestingly, homozygous *tmem43* mutant zebrafish exhibit normal heart morphology and contractility at early developmental stages, however, they develop late-onset enlarged hearts comparable to the overexpression models. These results indicate that aberrant TMEM43 contributes to the development of structural and functional cardiac defects through structural remodeling and transcriptomic alterations within the myocardium. 

By phylogenetic analysis, we first confirmed the presence of a single, corresponding *tmem43* ortholog in the zebrafish genome and, subsequently, analyzed its spatial and temporal expression pattern performing WISH and RT-PCR (Appendix A). Protein sequence alignment of TMEM43 reveals an overall high conservation among investigated vertebrate species. Remarkably, amino acid position p.S358 is situated in TMD3 and is conserved in all investigated species, thereby suggesting the importance of this residue for TMEM43 function (Appendix A). In contrast, the p.P111 position is not conserved in all investigated species, and alters in all non-mammalian species. The *tmem43* gene is expressed diffusely during embryonic development in the anterior region of the zebrafish, including the brain/head, the heart, and the otic vesicle. In addition, *tmem43* is transcribed ubiquitously in a broad range of adult zebrafish tissues similar to the expression pattern found in human organs. Of note, contrary to previous findings in humans, *tmem43* expression can be detected in zebrafish skeletal muscle [28]. Thus, the structural and transcriptional conservation of *tmem43* offers the possibility to use the zebrafish as a disease model system to gain further insights into the functional role of Tmem43 in the heart.

In addition to the TMEM43-p.S358L variant, which has been unequivocally confirmed causal for ACM, several missense variants at position p.P111 (p.P111A, p.P111Q, p.P111L) have also been linked to cardiac phenotypes (ClinVar database, April/2022) [24], but their relevance for human diseases remains uncertain. Those variants are mostly predicted to be tolerated with no clear prediction of pathogenicity by the different programs (Appendix A). The predicted neutral effects of the substituents are further demonstrated by computer simulations of TMEM43 protein structure (Figure 1). When replacing p.P111 with either A, Q, or L, no significant changes in protein structure were predicted. In contrast, at the interface between TMD3 and TMD4, the expanded side chain volume of p.S358L results in a steric conflict with the opposite residue p.I387 provoking a conformational change of TMEM43. Previous studies reported that pathogenic mutations in multi-spanning TMD proteins result in a compromised membrane topology and are accompanied by alterations in energetics of binding and folding of the TMD α-helices [43,44]. Moreover, it has been shown that already slight changes in energetics may affect the conformational stability [44]. Although, the substitution of serine to leucine might have only a minor effect on energetics, it might be sufficient to impair correct membrane topology of TMEM43, resulting in a disturbed ability to interact with other nuclear envelope or LINC complex proteins [22]. Furthermore, distorted TMEM43 appears to be destabilized, which is illustrated by the partial redistribution of TMEM43-p.S358L to the cytoplasm (Figure 2D,E). This agrees with findings in a transgenic mouse model, showing partially mislocalized TMEM43 in the cytoplasm of TMEM43-p.S358L overexpressing mice [30].

We assessed the effect of mutated human *TMEM43* on cardiac phenotype and ventricular contractility at 3 dpf. Embryos of both TMEM43 transgenic variants displayed an increased proportion of developmental defects, such as incorrect looping and pericardial edema, suggesting a pivotal role of TMEM43 for proper cardiogenesis (Figure 3A,B). The importance of nuclear membrane proteins for cardiac development has recently been reported in mice, as deletion of LEMD2 caused reduced expression of developmental genes and increased CM death, subsequently leading to an underdeveloped fetal heart [45,46]. Nevertheless, the morphological defects in the *TMEM43* expressing zebrafish were relatively mild, so that survival was not significantly affected. Interestingly, opposite to previous results, we measured increased ventricular systolic function in the TMEM43-S358L variant (Figure 3D) [30,34]. Cardiac hypercontractility may be caused by altered Ca^2+^ handling and/or disturbed sarcomeric mechanical properties [47,48,49]. The increased ventricular contractility in the TMEM43-S358L embryos might be a compensatory mechanism to preserve cardiac output. However, further experiments are necessary to confirm the observed alterations and to explore the underlying mechanism, for example by calcium or contractile force measurements at single-cell level.

We observed that cardiac-restricted overexpression of TMEM43-WT and TMEM43-p.P111L results in hypertrophic embryonic ventricles without compromising heart function or tissue structure, and this persists into adulthood (Figure 4C and Figure 5A–C). Furthermore, we determined that organ hypertrophy was mediated by enlarged CM size, but not by elevated proliferative capacity (Figure 4). Overexpression of a WT gene product can cause a defective phenotype, such as overgrowth of an organ by hyperplasia or hypertrophy [50,51,52]. We hypothesize that hyperactivation of the mTOR pathway and increased expression of genes involved in ribosome biogenesis, as revealed by transcriptomic analyses, contribute to the hypertrophic phenotype in the transgenic zebrafish (Figure 5D,E) [39]. Since this was not detected in the transgenic TMEM43 mouse model, it remains elusive whether the differences between mice and zebrafish result from the different onset of transgene expression at varying time points (the *myl7* (previously named *cmcl2*) promoter is active from 13-somite stage (14 hpf) onwards, whereas the murine alpha myosin heavy chain 6 (*αMHC6*) promoter becomes active in a late stage of cardiac development), or from other intrinsic mechanisms that differ between murine and zebrafish CMs [53,54].

The most common clinical manifestation of ACM is ventricular arrhythmia that include ventricular tachycardia or fibrillation and, in the worst case, result in cardiac arrest [3,55,56]. Although electrical conduction defects are related to remodeling of the intercalated disc structure, and/or alterations in Ca^2+^ homeostasis in early disease stage, a major driver of arrhythmias in the advanced end-stage is the fibro-fatty replacement of CMs within the myocardium [57,58,59,60]. Consistent with other genetic zebrafish lines modeling cardiomyopathies, we could not detect adipocytes or fibrotic tissue in the ventricles of adult zebrafish, which could be one reason for the absence of electrophysiological changes (Figure 5A and Appendix A) [61,62]. Another reason may be the lack of environmental factors that are supposed to contribute to the disease progression. Cardiac mechanical stress induced by training is an important factor that increases the risk of arrhythmias and heart failure in humans with ACM [63,64,65]. Thus, long-term physical exercise could deteriorate the cardiac phenotype in the transgenic zebrafish, and this remains to be tested in future studies.

Ultrastructural analysis of adult ventricular myocardial tissue indicates that changes in *TMEM43* expression in zebrafish are associated with ultrastructural alterations, such as altered sarcomere structure and mitochondrial morphology (Figure 6). However, despite the fact that TMEM43 is a nuclear envelope protein and is important for the maintenance of nuclear membrane structure and morphology, overexpression of TMEM43 variants apparently does not affect nuclear integrity in adult CMs. Beyond energy production, mitochondria provide important components for fatty acid and cholesterol synthesis, glucose and heme synthesis, and nucleotide and amino acid synthesis [66]. Perturbations in mitochondrial morphology have been associated with metabolic diseases, but also with dilated cardiomyopathy and heart failure [67,68,69]. Ultrastructural analysis revealed significantly reduced mitochondria size in both transgenic lines expressing TMEM43 variants, as well as altered shape in TMEM43-S358L compared to the NTG control, which may indicate metabolic changes. These findings are in accordance with the transcriptomic data, demonstrating that metabolic pathways are significantly enriched in both TMEM43 variant expressing lines and that the corresponding DEGs were mainly down-regulated (Figure 6F,G).

Interestingly, no significantly enriched KEGG pathways were found when comparing the DEGs of TMEM43-S358L with those of TMEM43-WT or NTG, suggesting that gene expression patterns do not differ essentially between those genotypes. We suspect that the low transcript amount of TMEM43-p.S358L accounts for the only minor differences in the cardiac phenotype and, subsequently, in the transcriptomes in this line. We cannot distinguish whether the protein level is low because the defective mRNA is degraded by NMD or whether fewer copies of the transgene were integrated into the zebrafish genome when the line was established.

Homozygous loss of function of Tmem43 in zebrafish has no effect on cardiac development or function in embryos, but leads to a late-onset heart enlargement (Figure 7). These findings agree with a previous study, reporting that heterozygous deletion of *Tmem43* under control of the *Myh6* promoter result in a late-onset senescence associated cardiac phenotype in mice [32]. In contrast, global *Tmem43* null mice displayed no overt cardiac dysfunction or morphological defects at baseline conditions or in response to pressure overload, indicating that TMEM43 is dispensable for murine cardiac development [31]. These differences indicate an ambiguous role of TMEM43 in murine cardiac development and function. Effects induced by the loss of *tmem43* might be masked by compensatory regulation of other nuclear envelope factors. Characterization of our zebrafish model, including analysis on molecular level, will help shed light on the role of *tmem43*.

The phenotype in the TMEM43-p.S358L overexpressing zebrafish line was milder compared to the phenotype described in the transgenic mice, as reflected by the absence of myocardial fibrosis and electrical defects in adult zebrafish [30]. This discrepancy might be explained by a different expression level of TMEM43-p.S358L and by intrinsic differences between the model species. Although we were not able to fully recapitulate the human ACM-phenotype with the present study, we demonstrate that tissue-specific overexpression of TMEM43-p.S358L and TMEM43-p.P111L cause ultrastructural and transcriptional changes in zebrafish hearts. These data provide further evidence for the essential role of *TMEM43* as disease gene for ACM. Nevertheless, it remains difficult to classify the TMEM43-p.P111L variant as pathogenic or benign, as the results presented here do not allow any clear conclusions to be drawn. However, our data suggest an influence of TMEM43-p.P111L on cardiac development, but this might be a zebrafish specific effect. In conclusion, during embryonic development we propose that overexpression of TMEM43 variants affects embryonic heart development, whereas the general reduction in Tmem43 function in CRISPR mutants is apparently well tolerated in zebrafish larvae. In adult fish, loss and gain of function of *tmem43*/*TMEM43* appear to have an effect on heart size, although the detailed reasons require further investigation.

## 4. Materials and Methods

### 4.1. Zebrafish Husbandry

Zebrafish (*Danio rerio*) are maintained, raised, and staged as previously described [1,2] in the aquatic facilities of the CHFC of the University Hospital Würzburg, Germany according to the Federation of European Laboratory Animal Science Associations guidelines. Adult zebrafish are kept at a constant temperature of 28 °C in 3.5 L plastic tanks (ZebTEC, Techniplast Deutschland, Hohenpeißenberg, Germany) with a daily cycle of 14 h light/10 h dark. Reverse osmosis water with an adjusted conductivity of 300–500 µS/cm and a pH of 7.5 is used. Animals are fed twice daily with *Artemia nauplii* and/or GEMMA Micro Food (size depends on the age; Skretting, Tooele, UT, USA). Embryos younger than 5 dpf are kept in E3 medium (5 mM NaCl, 0.17 mM KCl, 0.33 mM CaCl_2_ and 0.33 mM MgSO_4_) in an incubator at a temperature of 28 °C. All experimental procedures were carried out in accordance with the guidelines of the German animal welfare law and were approved by the local government of Lower Franconia (Tierschutzgesetz §8, Abs. 1; approval number.: 55.2.2-2532-2-355, 55.2.2-2532-2-602). In all studies the zebrafish strain Tüpfel long fin (ZDB-GENO-990623-2) was used. Phenylthiourea (PTU; Sigma-Aldrich, St. Louis, MO, USA) was added to E3 raising medium (0.003% *v*/*v*) starting from stage 24 hpf on to inhibit pigmentation for in vivo imaging, confocal analysis and in situ hybridization.

### 4.2. Generation of Transgenic Lines

To generate transgenic zebrafish lines with cardiac-specific overexpression of *TMEM43*, Gateway technology (Invitrogen, Waltham, MA, USA) according to Kwan et al. [70] was applied by cloning eGFP-C2-*TMEM43* fragments with either one of the two variants of interest (c.1073C>T, p.S358L; c.332C>T, p.P111L) and the eGFP-C2-*TMEM43* wild-type fragment downstream to the zebrafish *myl7* promoter into the Tol2kit expression system, which was kindly provided by Prof. Koichi Kawakami. For in vitro synthesis of capped transposase mRNA, pCS2FA-transposase plasmid (Addgene #133032) [71] was linearized with *Not*I (Thermo Fisher Scientific, Waltham, MA, USA) and transcribed using the mMESSAGE mMACHINE SP6 Transcription Kit (Invitrogen, USA). eGFP-C2-*TMEM43* Tol2 expression constructs together with *tol2* transposase mRNA were co-injected into zebrafish embryos to generate transgenic founder (F0) lines. Progeny from transgenic F2 in-cross mating were used for all analyses within this manuscript. The generated transgenic lines TMEM43-WT (ca47Tg; ZFIN ID: ZDB-ALT-220722-2), TMEM43-P111L (ca46Tg; ZFIN ID: ZDB-ALT-220725-1), and TMEM43-S358L (ca44Tg; ZFIN ID: ZDB-ALT-220725-2) were submitted to ZFIN.org.

### 4.3. Generation of tmem43 CRISPR Lines

sgRNAs were designed and generated as described previously [40]. Briefly, the ZiFit Targeter website (http://zifit.partners.org/ZiFiT/ChoiceMenu.aspx (accessed on 27 July 2017)) [41,42] was used to identify a suitable target site within *tmem43* (genome assembly Zv9: ENSDARE00000787164) meeting the following criteria: 5′-GG-(N)_18_-NGG-3′. sgRNA target site (GGCCGCCGGGCTTTCTGGAG) is located in exon 2 of *tmem43.*


The phosphorylated single-stranded oligonucleotides 5′-TAGGCCGCCGGGCTTTCTGGAG-3′ and 5′-AAACCTCCAGAAAGCCCGGCGG-3′ were annealed and subsequently cloned into the linearized DR274 (Addgene #42250) plasmid. sgRNAs were synthesized by linearization of the plasmid with *Dra*I (Thermo Fisher Scientific) and using the plasmid as template for the MAXIscript T7 Transcription Kit (Invitrogen). In vitro synthesis of capped *Cas9* mRNA was carried out by linearizing plasmid pCS-nCas9n (Addgene #47929) [72] with endonuclease enzyme *Not*I (Thermo Fisher Scientific) and using the linearized plasmid as template for the mMESSAGE mMACHINE SP6 Transcription Kit (Invitrogen). After linearization, plasmids were purified using the Qiaquick PCR Purification Kit (Qiagen, Hilden, Germany). The synthesized mRNAs were finally purified by phenol/chloroform extraction. The generated *tmem43^wue4/wue4^* (ZFIN ID: ZDB-ALT-220726-2), *tmem43^wue5/wue5^* (ZFIN ID: ZDB-ALT-220726-3) and *tmem43^wue6/wue6^* (ZFIN ID: ZDB-ALT-220726-4) lines were submitted to ZFIN.org.

### 4.4. Microinjection into Zebrafish Embryos

Zebrafish embryos were microinjected at one-cell stage using the FemtoJet 4i (Eppendorf, Hamburg, Germany) microinjector together with Femtotip II (Eppendorf, Germany) microcapillaries. The composition of the injection mixture depended on the experimental approach. In case of generating transgenic TMEM43 zebrafish lines, the injection mixture consisted of 25 ng/µL *tol2* transposase mRNA, 25 ng/µL Tol2 expression construct, 0.05% Phenol Red (pH 7.0; Carl-Roth, Karlsruhe, Germany) and 1 mg/µL FITC Dextran (Sigma-Aldrich). At 3 dpf, injected embryos were inspected under fluorescence stereomicroscope and screened for *GFP*-positive cells in the heart. In case of generating *tmem43* mutant lines, the injection mixture consisted of 100 ng/µl *Cas9* mRNA, 25 ng/µL *tmem43* exon2 sgRNA, 0.05% Phenol Red and 1 mg/µL FITC Dextran. At 24 hpf, injected embryos were analyzed under stereomicroscope for proper development and only healthy embryos were screened for target site mutations. DNA was extracted from 10 single embryos and sequenced for identification of CRISPR/Cas9 induced indel mutations.

### 4.5. Whole-Mount In Situ Hybridization

Riboprobe synthesis and whole-mount in situ hybridization (WISH) was performed following standard procedures as previously described [73]. Briefly, RNA probes for *tmem43* were generated as follows: a transcript fragment of *tmem43* (BC154204.1) was amplified by polymerase chain reaction (PCR) using zebrafish cDNA as template. Used primers tmem43-WISH_for: 5′-CGGAAGTGGAGTTAGCGTTAGTG-3′ and tmem43-WISH_rev: 5′-GCATGCACAGACACACTGTAG-3′, with an amplicon size of 357 bp. Quality and size of the PCR amplicon was checked by agarose gel electrophoresis. After purification using the Qiaquick PCR Purification Kit (Qiagen), the PCR product was cloned into pCRII vector (TA Cloning Dual Promoter Kit, Invitrogen) and verified by Sanger sequencing. In vitro transcription of the riboprobes was performed by linearizing the plasmid with appropriate restriction enzyme and transcribing with DIG-labeling mix and appropriate RNA polymerase. For *tmem43* antisense (working) probe, linearization was completed with *Not*I (New England Biolabs, Ipswich, MA, USA) and transcription with SP6 RNA polymerase (Roche Diagnostics, Rotkreuz, Switzerland). For *tmem43* sense (negative control) probe, linearization was performed with *Bam*HI (New England Biolabs, USA) and transcription with T7 RNA polymerase (Roche Diagnostics). WISH was performed on zebrafish embryos previously fixed at the desired developmental stage (24 hpf, 2 dpf, 3 dpf) with 4% paraformaldehyde (PFA) in PBS and stored in 100% methanol at −20 °C and were performed in triplicates with processing at least 10 embryos per condition in a single tube. 

### 4.6. In Vivo Imaging and Analysis

Video recording was conducted using a Leica Stereomicroscope M205 FA equipped with a DFC9000 GTC CCD camera (Leica Microsystems, Wetzlar, Germany) with 160-fold magnification at 250 frames per sec. Zebrafish embryos 3 dpf were anesthetized in 160 mg/L MS222 (Sigma-Aldrich) in E3 medium and placed horizontally in a small Petri dish filled with E3 medium to obtain lateral view. The left eye was facing downward for optimal imaging of the ventricle.

The GFP-positive transgenic embryos were visually phenotyped based on the recorded videos and the number of cardiac-specific defects was determined. The defects were classified as follows: looping defects, pericardial edemas, arrhythmias, altered ventricular shape, strong ventricular contractions, heterogeneous ventricular contractions, and altered atrial shape. 

Images of end-diastole and end-systole of three contractile cycles were analyzed with ImageJ/Fiji (https://fiji.sc/ (accessed on 14 March 2017)). To determine end-diastolic ventricular area and end-systolic ventricular area the perimeter of the ventricle was outlined. Length of the short axis (width) was measured between the myocardial borders of the ventricle to calculate FS as follows: %FS = [(width at diastole − width at systole)/(width at diastole)]. For %FS only embryos with proper looped hearts were analyzed.

### 4.7. Surface ECG Recording and Analysis

Adult zebrafish were anesthetized with MS222 in fish tank water (final concentration ~150 mg/L) and placed dorsal side down on a special pedestal with a small cavity. Before two electrodes were placed, conductive gel was applied. Non-invasive surface ECG recordings were performed using the ZS-200 zebrafish ECG recording system from iWorx (iWorx Systems, Dover, NH, USA) consisting of the IX-100F recorder, the A-ZEC-200 recording chamber and Ag/AgCl electrodes, and analyzed using the accompanying LabScribe v3 (iWorx Systems) analysis software. ECG signals were recorded for up to 4 minutes before the zebrafish were transferred to fresh fish tank water for recovery. Raw digitized signals were filtered to remove noise by applying a Barlett window-based FIR filter (low cut-off filter: 20 Hz; high cut-off filter: 500 Hz) and smoothed using the moving average. Poincaré plots were generated to demonstrate beat-to-beat variation by measuring RR intervals of 20 consecutive heartbeats at two different timepoints of the ECG trace. Heart rate was determined by counting heartbeat for 10 s. at four different time points, calculating the mean value and multiplying this number by six.

### 4.8. Immunofluorescence Staining 

For whole-mount immunofluorescence staining, PFA and methanol fixed embryos were rehydrated in a descending methanol series (75%, 50%, 25%, PBS; 5 min each, with two changes in PBS), permeabilized with 10 µg/ml proteinase K (Invitrogen) for 10 min at 65 °C and post-fixed in 4% PFA for 30 min at room temperature. After washing with PBS/Tween 20 and blocking (5% goat serum, 1% BSA, 0.5% Triton-X100 in PBS) for at least 2 h, embryos were incubated with primary antibody solutions, including S46 (1:10; Developmental Studies Hybridoma Bank, Iowa City, IA, USA), GFP (1:200; Santa Cruz Biotechnology, USA), and PCNA (1:400; Genetex, Irvine, CA, USA) at 4 °C overnight. The next day, embryos were washed with PBS/Triton-X100 and incubated with secondary antibody solutions goat anti-rabbit Alexa Fluor 488 (1:500; Invitrogen) and goat anti-mouse Alexa Fluor 594 (1:500; Invitrogen) at 4 °C overnight. Cell nuclei were counterstained with DAPI (Carl-Roth) and specimen were mounted in 1.2% low-melting agarose (Carl-Roth). For immunostaining with Alcam (1:100; Developmental Studies Hybridoma Bank), blocking was performed with 10% goat serum, 0.1% Tween 20 and 1% DMSO in PBS.

Immunofluorescence staining with pHH3 antibody on dissected embryonic hearts was carried out as previously described with minor modifications [74]. Briefly, embryos at the desired stage were euthanized by immersion in 300 mg/L MS222, hearts were dissected using two insulin syringes and transferred to slides with concave cavities. Hearts were fixed in 4% PFA for 20 min at room temperature, washed with PBS/Tween 20, and blocked with 10% goat serum in PBS/Tween 20 for 1 h at room temperature. Embryonic hearts were then incubated with primary antibody solutions pHH3 (1:400; Cell Signaling Technology, Danvers, MA, USA) and MF20 (1:10; Developmental Studies Hybridoma Bank) for 1 h at room temperature. After several washes with PBS/Tween 20, samples were incubated with secondary antibody solutions goat anti-rabbit Alexa Fluor 594 (1:500; Invitrogen) and goat anti-mouse Alexa Fluor 488 (1:500; Invitrogen). Cell nuclei were counterstained with DAPI and specimen were mounted in Mowiol (Sigma-Aldrich).

Immunofluorescence staining of adult dissected hearts was performed on 5 µm paraffin sections. Antigen retrieval was carried out by boiling the sections in sodium citrate buffer (10 mM Tri-sodium citrate dihydrate; pH 9.0) for 10 min. After washing with PBS, the sections were incubated with 0.3% H_2_O_2_ in methanol for 10 min, then in 50% ethanol for 2 min and permeabilized with 0.2% Nonidet (United States Biological, Salem, MA, USA) in PBS for 5 min. The sections were blocked with 10% goat serum in PBS for at least 2 h and antibody staining with GFP (1:200; Santa Cruz Biotechnology, Dallas, TX, USA) was performed in blocking solution at 4 °C overnight. Washing was carried out with PBS and secondary antibody staining used Alexa Fluor 488 goat anti-rabbit (1:500; Invitrogen) in blocking solution. Cell nuclei were counterstained with DAPI. Confocal z-stacks were acquired using a Zeiss LSM 780 confocal microscope (Carl Zeiss Microscopy, Jena, Germany). Image processing of all immunofluorescence stainings were performed using the ImageJ/Fiji software.

### 4.9. Histologic Analyses

Histology of embryonic zebrafish was performed as previously described [75]. Embryos at 3 dpf were anesthetized by immersion in MS222 solution, fixed in 4% PFA and embedded in JB-4 plastic (Polysciences, Warrington, PA, USA) according to manufacturer’s protocol with correct orientation for sagittal sections. The plastic sections were stained with H&E (Carl-Roth). 

Adult zebrafish were euthanized by immersion in a lethal MS222 solution (final concentration ~300 mg/L) and heart was excised after loss of righting reflexes. Dissected hearts were fixed in 4% methanol-free PFA (Polysciences) at 4 °C overnight, embedded in paraffin wax and sectioned. The heart was orientated for sagittal sections through the ventricle, atrium and *Bulbus arteriosus*. H&E and MTC (Sigma-Aldrich) staining were completed according to manufacturer’s protocols as previously described [76]. Sectioning of the resin and paraffin blocks at 5 µm were performed on a Leica RM 2245 vibratome (Leica Biosystems, Nußloch, Germany). All histologic images were captured on a Keyence Biozero BZ-8000K microscope (Keyence, Osaka, Japan).

### 4.10. Transmission Electron Microscopy

The 5-month-old zebrafish were euthanized by immersion in a lethal MS222 solution and hearts were excised after loss of righting reflexes. Chemical fixation and sectioning was performed by the Imaging Core Facility at the University of Würzburg as previously described [77]. Ultrathin sections were examined with a JEOL JEM-2100 (Jeol, Freising, Germany) or a JEOL JEM-1400 Flash (Jeol) and digital images were taken with a TemCam F416 high-resolution CMOS camera (Tietz Video and Imaging Processing Systems, Gauting, Germany) or a Mataki flash sCMOS camera (Jeol).

### 4.11. Sanger Sequencing

For amplification of the genomic region of interest, target site-specific PCR primers were designed using Primer Blast (NIH) and a touchdown PCR in combination with a Mango *Taq* Polymerase (Bioline, London, UK) was performed. The annealing temperature was adjusted according to the melting temperatures of the primers. TMEM43-specific point mutations were confirmed using following primers, eGFP_for: 5′-CATGGTCCTGCTGGAGTTCGTG-3′ and TMEM43-c.332_rev: 5′-TGGGATTTTCGCTGTGGTAGAA-3′, with an amplicon size of 797 bp; TMEM43-c.1073_for: 5′-TTCTACCACAGCGAAAATCCCA-3′ and TMEM43-c.1073_rev: 5′-CGTGTCCGAGCAACAAGGATGG-3′, with an amplicon size of 506 bp. The following primers were used to study *tmem43* CRISPR mutations, tmem43-CRISPR_for: 5′-TGTGATTAGCGCGGGTTTCT-3′ and tmem43-CRISPR_rev: 5′-TCGCTGTACTCCACCCATTG-3′, with an amplicon size of 791 bp. PCR products were cleaned up using the ExoSAP-IT PCR Product Cleanup kit (Thermo Fisher Scientific) and sequenced by Microsynth Seqlab (Balgach, Switzerland) or Eurofins Genomics (Luxembourg). Sequencing traces were visualized and analyzed with SnapGene (San Diego, CA, USA).

### 4.12. RT-PCR

Total RNA was extracted from a pool of wild-type zebrafish embryos at 24 hpf, 2 and 3 dpf, and from adult organs (heart, liver eye, ovary/testis, brain, and skeletal muscle) using Qiazol (Qiagen) according to manufacturer’s instructions. For removal of residual DNA, the extracted RNA was incubated with DNAse I (Sigma-Aldrich) for 15 min at room temperature. A total of 2 µg of total RNA was reverse-transcribed using the High-Capacity RNA-to-cDNA Kit (Applied Biosystems, Waltham, MA, USA) and used as template for RT-PCR analysis with *MangoTaq* DNA Polymerase (Bioline) and *eef1a1l1* (NM_131263.1) as reference gene. Primers for *tmem43* and *eef1a1l1* were tmem43-WISH_for: 5′-CGGAAGTGGAGTTAGCGTTAGTG-3′ and tmem43-WISH_rev: 5′-GCATGCACAGACACACTGTAG-3′, with an amplicon size of 357 bp; eef1a1l1_for: 5′-GCCCCTGGACACAGAGACTTCATCA-3′ and eef1a1l1-_rev 5′-ATGGGGGCTCGGTGGAGTCCAT-3′, with an amplicon size of 211 bp. PCR products were agarose-gel separated, stained with Midori Green (Nippon Genetics Europe, Düren, Germany) and imaged under UV light using a PXi imaging system (Alpha Metrix Biotech, Rödermark, Germany). Amplicon size was compared with a 100 bp DNA ladder (Thermo Fisher Scientific).

### 4.13. qPCR

Dissected ventricular or skeletal muscle tissue of adult zebrafish was homogenized with a Sonoplus Mini20 homogenizer (Bandelin Electronic, Berlin, Germany) and total RNA was extracted using Qiazol (Qiagen) according to manufacturer’s instructions. For removal of residual DNA, the extracted RNA was incubated with DNAse I (Sigma-Aldrich) for 15 min at room temperature. Up to 300 ng total RNA was then reverse-transcribed using the iScript cDNA Synthesis Kit (Bio-Rad Laboratories, Hercules, CA, USA). Transcript level of *TMEM43* and *tmem43* was analyzed by performing TaqMan Gene Expression Assay (Thermo Fisher Scientific) according to manufacturer’s instruction. The following TaqMan probes were used, *TMEM43*: Hs00225702_m1, *tmem43*: Dr03101329_m1 and *eef1a1l1*: Dr03432748_m1 as internal control. The PCR program for amplification was 95 °C for 14 min, followed by 45 cycles at 95 °C for 15 s, 60 °C for 35 s, and 72 °C for 25 s. All qRT-PCRs reactions were performed in duplicates with a CFX96 Real-Time PCR Detection System (Bio-Rad Laboratories).

### 4.14. RNA-Seq

Total RNA was extracted from excised ventricular tissue of adult *TMEM43* overexpressing zebrafish and non-transgenic controls using Qiazol (Qiagen) according to manufacturer’s protocol. Up to seven ventricles were pooled as one sample and three biological replicates for each genotype were analyzed. RNA quality check, library preparation, and sequencing of libraries were performed by the Core Unit Systems Medicine at the University of Würzburg. DNA libraries were prepared using the TruSeq Stranded mRNA Library Preparation Kit (Illumina, San Diego, CA, USA) according to manufacturer’s instructions with 150 ng of total RNA and 15 PCR cycles. All libraries were pooled and sequenced with 25 million reads/sample in single-end mode with 75 nt read length on the NextSeq 500 platform (Illumina). For ensuring high sequence quality, Illumina reads were quality- and adapter-trimmed via Cutadapt [78] version 2.5 using a cutoff Phred score of 20. Further, processed reads were aligned to the Zebrafish genome version GRCz11 (Genome Reference Consortium Zebrafish Build 11) using STAR v2.7.2b [79]. Reads mapped to the exome were quantified via featureCounts v1.6.4 [80] and determined read counts served as input to DESeq2 version 1.24.0 [81] for identification of differentially expresses genes. Only genes with a Benjamini–Hochberg corrected adjusted *p*-value (padj) < 0.05 and |log2FoldChange| ≥ 0.05 were assumed to be significantly differentially expressed. ShinyGO v0.76 was used to perform functional enrichment analyses based on KEGG pathways (http://bioinformatics.sdstate.edu/go/ (accessed on 27 April 2022)) [82]. Clustered heatmaps were drawn using the pheatmap R package with normalization of the values by rows.

### 4.15. Western Blot Analysis

From excised hearts of 5-month-old zebrafish, the atrium and *Bulbus arteriosus* were removed and only the ventricles were used for sample preparation. Single or a pool of ventricles were homogenized in lysis buffer (300 mM NaCl, 100 mM NaH_2_PO_4_ × H_2_O, 50 mM Na_2_HPO_4_, 1 mM MgCl × 6 H_2_O, 10 mM Na_4_P_2_O_7_, 12.7 mM EDTA, 2.7 mM NaF, 1 mM sodium orthovanadate) in presence of protease inhibitor cocktail (Roche Diagnostics) and 1 mM PMSF (Sigma-Aldrich) by sonication on ice. Homogenized tissues were centrifuged at 1500× *g* at 4 °C for 15 min, supernatants were collected as samples, mixed with Laemmli buffer (Bio-Rad Laboratories) and incubated at 95 °C for 5 min. Proteins were separated by SDS-PAGE using Mini-PROTEAN TGX precast protein gel (Bio-Rad Laboratories) and transferred to polyvinylidene fluoride membrane using the Trans-Blot Turbo Transfer System (Bio-Rad Laboratories). After blocking with 5% non-fat milk in PBS/Tween 20, the membrane was incubated with primary antibodies, including GFP (1:200; Santa Cruz Biotechnology, USA), TMEM43 (1:250; Sigma-Aldrich), and Vinculin (1:2000; Abcam, Cambridge, UK) at 4 °C overnight and incubated with horseradish peroxidase-conjugated secondary antibodies (goat anti-rabbit IgG: 1:1000, GE Healthcare, USA; goat anti-mouse IgG: 1:1000, GE Healthcare, USA) for 2 h at room temperature. Signals were detected by luminol-based chemiluminescence using ECL prime reagent (GE Healthcare, Chicago, IL, USA) and a ChemiDoc Touch Imaging System (Bio-Rad Laboratories). Intensity of the protein bands were quantified by ImageJ/Fiji. 

### 4.16. Bioinformatical Analysis

Evolutionary conservation of the TMEM43 protein across species was determined using Clustal Omega [83]. The potential effect of amino acid substitutions on TMEM43 protein structure were predicted with AlphaFold-2 [84] and illustrated using the PyMOL Molecular Graphics System (Schrödinger, New York, NY, USA). Variants were classified using the different algorithms of BayesDel-addF, BayesDel-noAF, PON-P2, ClinPred, and Poly-Phen-2 [36,85,86,87]. Evaluation of allele frequencies was performed using the Genome Aggregation Database v2.1.1 (gnomAD; https://gnomad.broadinstitute.org/ (accessed on 2 April 2022)). 

### 4.17. Statistical Analysis

Statistical analysis was carried out using Prism 9 (GraphPad Software, San Diego, CA, USA). Data are shown as mean ± standard error of the mean (SEM). For analyzing differences among two groups, unpaired Student’s *t*-test, and for comparison between three or more groups one-way ANOVA followed by Bonferroni’s was used. A *p*-value of ≤ 0.05 was considered significant; * *p* ≤ 0.05, ** *p* ≤ 0.01, *** *p* ≤ 0.001; **** *p* ≤ 0.0001. Data from transgenic lines were compared with the NTG control, in addition data from both TMEM43 variants were compared with the TMEM43-WT. Homozygous *tmem43* mutant zebrafish were compared to WT and heterozygous *tmem43* mutant zebrafish.

## Figures and Tables

**Figure 1 ijms-23-09530-f001:**
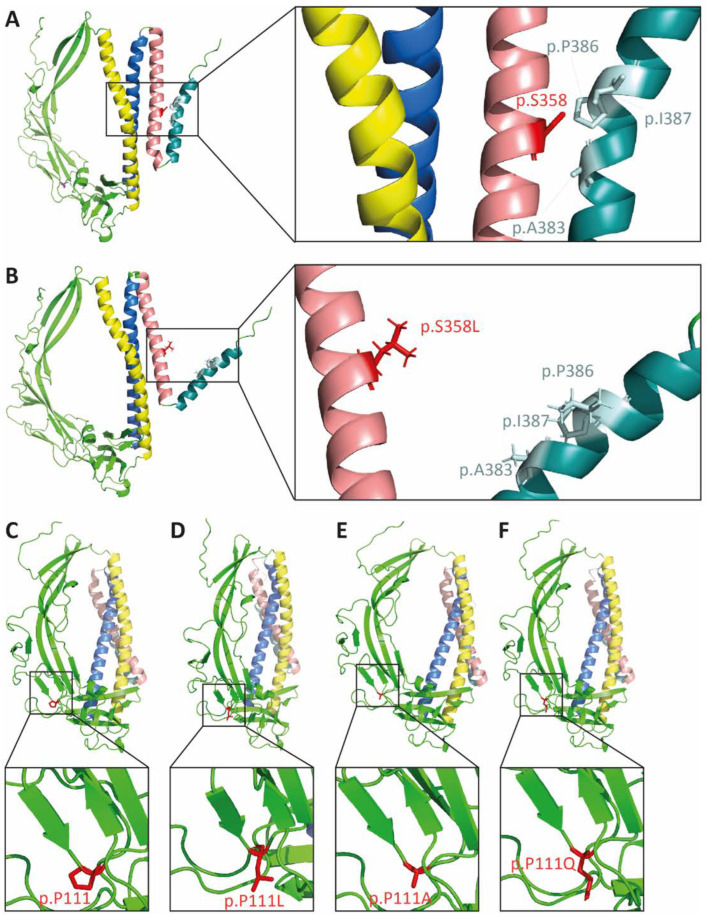
The p.S358L variant changes protein conformation of TMEM43. In silico modeling of TMEM43 tertiary structure of wild-type (**A**,**C**) and proteins with the corresponding missense mutations (**B**) p.S358L, (**D**) p.P111L, (**E**) p.P111A, and (**F**) p.P111Q. Yellow indicates transmembrane domain (TMD) 1, blue indicates TMD2, rose indicates TMD3, and turquoise indicates TMD4. The residue of interest is indicated in red (p.S358 in **A** and **B**; p.P111 in **C**–**F**).

**Figure 2 ijms-23-09530-f002:**
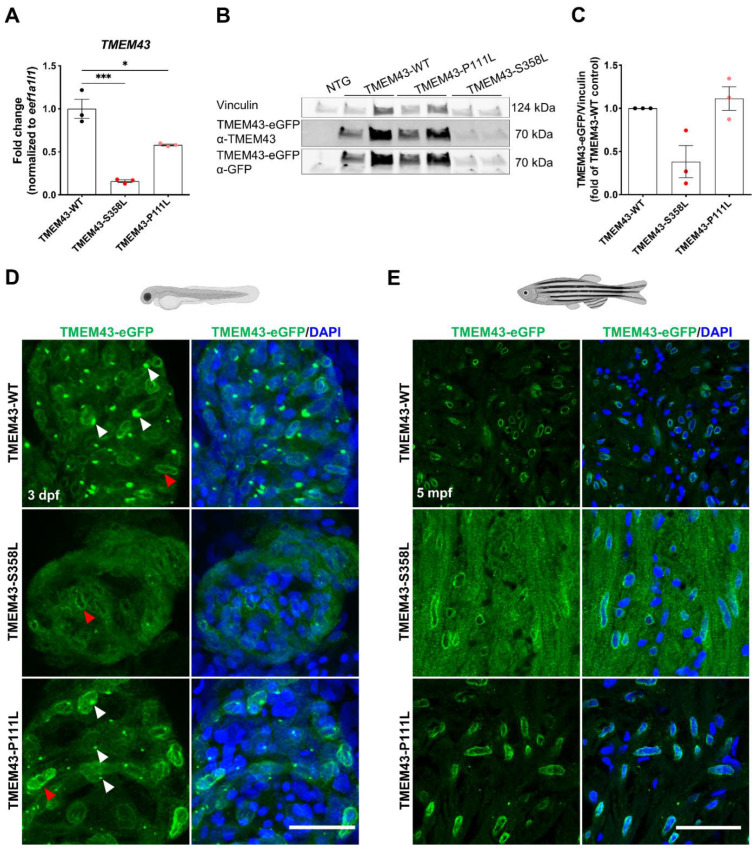
Reduced expression and altered localization of TMEM43-p.S358L. (**A**) Quantitative reverse transcription PCR demonstrates transcript reduction in human TMEM43 in adult ventricles of both TMEM43 variant transgenic lines compared to wild-type (WT). Each replicate consists of a pool of 2–7 ventricles and relative expression is calculated to TMEM43-WT expression levels. (**B**) Representative Western blot showing reduced occurrence of the human TMEM43-eGFP fusion-protein in adult ventricles of TMEM43-S358L compared to the TMEM43-WT transgenic line. For detection, both a TMEM43- and a GFP-antibody were used. Each biological replicate consists of a pool of 1–4 ventricles. (**C**) Western blot analysis using the TMEM43-antibody showed a reduced protein level of the human TMEM43-eGFP fusion-protein in adult ventricles of TMEM43-S358L compared to the TMEM43-WT transgenic line. Each data point represents relative Western blot band intensity of a technical replicate. (**A**,**C**) One-way ANOVA with Bonferroni’s multiple comparison test, * *p* ≤ 0.05, *** *p* ≤ 0.001. Error bars correspond to SEM. (**D**) Confocal images of whole-mount immunofluorescence staining of hearts expressing the transgene TMEM43-eGFP (green) at 3 days post-fertilization (dpf). Note the TMEM43-WT expression at the nuclear membrane and the endoplasmic reticulum (ER). TMEM43-S358L displays a weak signal at the nuclear membrane, but no signal at the ER. White arrowheads indicate TMEM43-eGFP localization at the ER, red arrowheads indicate localization at the nuclear membrane. Scale bar = 30 µm. (**E**) Confocal images of immuno-stained paraffin sections of 5 months post fertilization (mpf) dissected hearts expressing the transgene TMEM43-eGFP (green). Scale bar = 30 µm. Created with BioRender.com (accessed on 13 August 2022).

**Figure 3 ijms-23-09530-f003:**
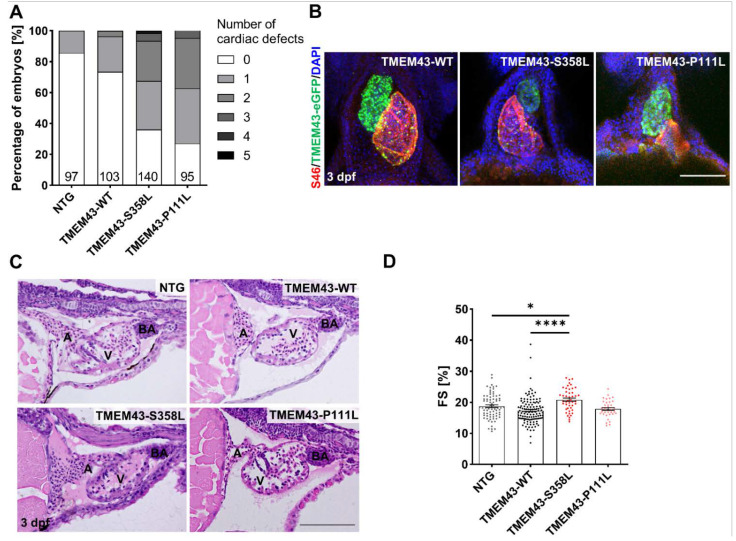
Overexpression of TMEM43 variants induce altered cardiac morphology and function at 3 days post-fertilization (dpf). (**A**) Visual phenotyping demonstrates the percentage of embryos with zero to five different cardiac-specific defects. The defects were classified into looping defects, pericardial edemas, arrhythmias, altered ventricular shape, strong ventricular contractions, heterogenous ventricular contractions, and altered atrial shape. Both TMEM43 variant lines show an increased percentage of embryos with at least one cardiac defect. The number of phenotyped embryos is indicated at the bottom of the columns. (**B**) Confocal maximum intensity projections of immunofluorescence-stained hearts. Atria are labeled for S46 localization (red), ventricles for GFP localization (green), and nuclei with DAPI (blue). In contrast to TMEM43-WT hearts, both TMEM43 variant lines display developmental defects (looping defects and reduced ventricular chamber size in TMEM43-S358L, as well as altered atrial chamber shape in TMEM43-P111L). Ventral views, arterial pole to the top. Scale bar = 100 µm. (**C**) Representative histological images of hematoxylin and eosin-stained sagittal plastic sections showing structurally normal heart chambers and the outflow tract in embryos at 3 dpf (*n* ≥ 5 per genotype). A, atrium; BA, Bulbus arteriosus; V, ventricle. Scale bar = 100 µm. (**D**) Analysis of cardiac function by ventricular fractional shortening (FS) quantification from a time series of the embryonic cardiac cycle. Data points represent the average measurement obtained from three diastolic/systolic phases of each fish. The short axis of the ventricle was used for quantification. Significantly increased FS was detected in TMEM43-S358L embryos. Significance was determined by one-way ANOVA with Bonferroni’s multiple comparison test, * *p* ≤ 0.05, **** *p* ≤ 0.0001. Error bars correspond to SEM. (**A**,**D**) Analyses were performed on embryos from three independent clutches per genotype.

**Figure 4 ijms-23-09530-f004:**
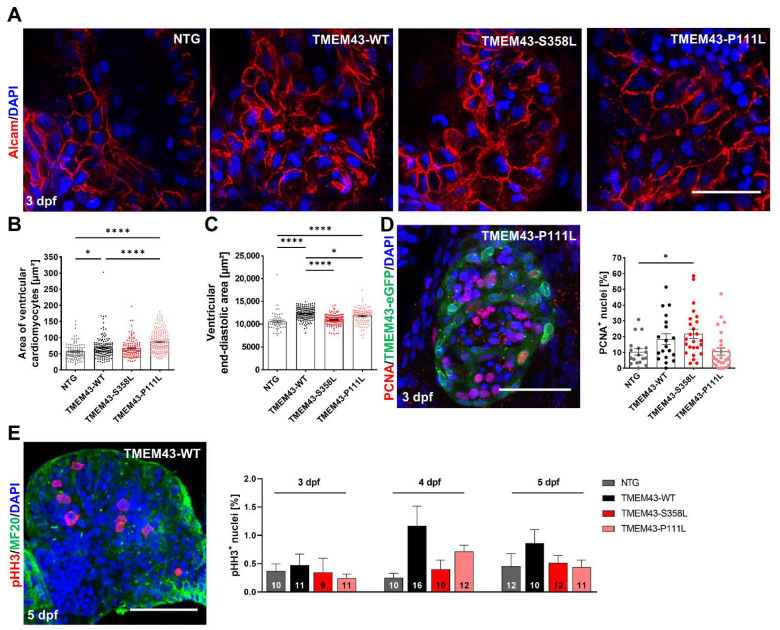
Cardiac overexpression of stably folded TMEM43 leads to hypertrophic ventricular growth, but unchanged proliferation capacity in embryos. (**A**) Confocal maximum intensity projections of 3 days post fertilization (dpf) hearts immunostained with DAPI (blue) and Alcam (red) to highlight plasma membranes. Scale bar = 30 µm. (**B**) Quantification of cardiomyocyte (CM) area represented in (**A**) reveals larger CMs in TMEM43-WT and TMEM43-P111L compared to non-transgenic control (NTG) and TMEM43-S358L individuals (n ≥ 10 hearts per genotype). (**C**) Quantification of ventricular end-diastolic area shows the mean area of three individual sections of each fish from a time series of the embryonic cardiac cycle. Significantly increased end-diastolic ventricular areas were measured in TMEM43-WT and TMEM43-P111L compared to NTG and TMEM43-S358L individuals. (**D**) Representative confocal maximum intensity projection of a heart immunostained with PCNA (red), GFP (green), and DAPI (blue) at 3 dpf. Scale bar = 50 µm. In the graph, individual data points represent the percentage of PCNA^+^ nuclei in the ventricle of each fish (n ≥ 18 individuals per genotype). A slightly increased proliferation rate was observed in the TMEM43-S358L transgenic line compared to NTG. (**E**) Representative maximum intensity projection of a dissected heart immunostained with pHH3 (red), MF20 (targeting MYH1E; green) and DAPI (blue) at 5 dpf. Scale bar = 50 µm. The graph displays the mean percentage of pHH3^+^ cell nuclei for each genotype at 3 dpf, 4 dpf, and 5 dpf. The number of analyzed hearts per condition are indicated at the bottom of the columns. (**B**–**E**) For all graphs, significance was determined by one-way ANOVA with Bonferroni’s multiple comparison test, * *p* ≤ 0.05, **** *p* ≤ 0.0001. Error bars correspond to SEM.

**Figure 5 ijms-23-09530-f005:**
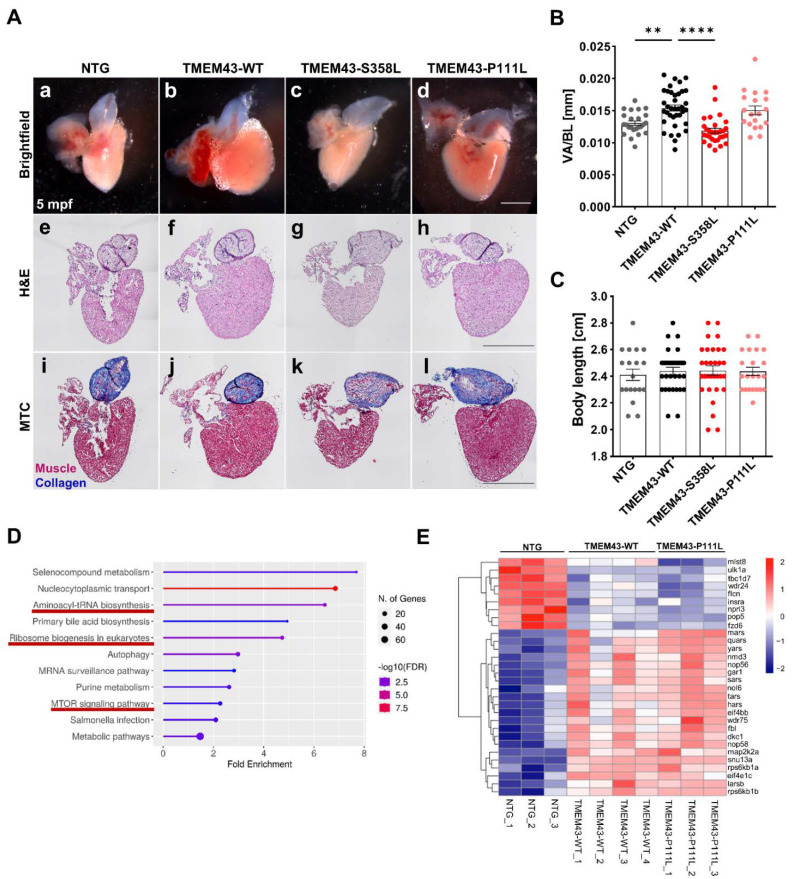
TMEM43 wild-type (WT) and TMEM43-P111L transgenic lines display enlarged hearts in adult zebrafish. (**A**) Preserved myocardial structure in adult zebrafish at 5 months post-fertilization (mpf) (*n* ≥ 5 hearts per genotype). Scale bars = 500 µm. (**a**–**d**) Representative brightfield images of dissected fish hearts. (**e**–**h**) Histology was analyzed by hematoxylin and eosin (H&E) staining. (**i**–**l**) Fibrosis was analyzed by Masson’s Trichrome (MTC) staining. (**B**) TMEM43-WT and TMEM43-P111L individuals display larger ventricles as indicated by an increased ventricular area to body length ratio (VA/BL) compared to the non-transgenic control (NTG) and TMEM43-S358L. Brightfield images represented in (**a**–**d**) were used for assessment of ventricular surface area. (**C**) Quantification of body length reveals normal growth and development of adult transgenic zebrafish. (**D**) Kyoto Encyclopedia of Genes and Genomes (KEGG) pathway enrichment analysis for shared differently expressed genes (DEGs) between “NTG vs. TMEM43-WT” and “NTG vs. TMEM43-P111L” (p_adj_ ≤ 0.05) using ShinyGO v0.76. The x-axis represents fold enrichment and y-axis represents enriched KEGG terms. The size of the dots represents the number of involved genes. The false discovery rate (FDR) value is indicated by colors.KEGG pathways, which include the DEGs shown in (**E**), are underlined in red and play a role for cell growth. (**E**) Heatmap showing DEGs belonging to the KEGG pathways “Aminoacyl-t-RNA biosynthesis”, “Ribosome biogenesis in eukaryotes”, and “mTOR signaling pathways”. (**B**,**C**) For all graphs, significance was determined by one-way ANOVA with Bonferroni’s multiple comparison test, ** *p* ≤ 0.01, **** *p* ≤ 0.0001. Error bars correspond to SEM.

**Figure 6 ijms-23-09530-f006:**
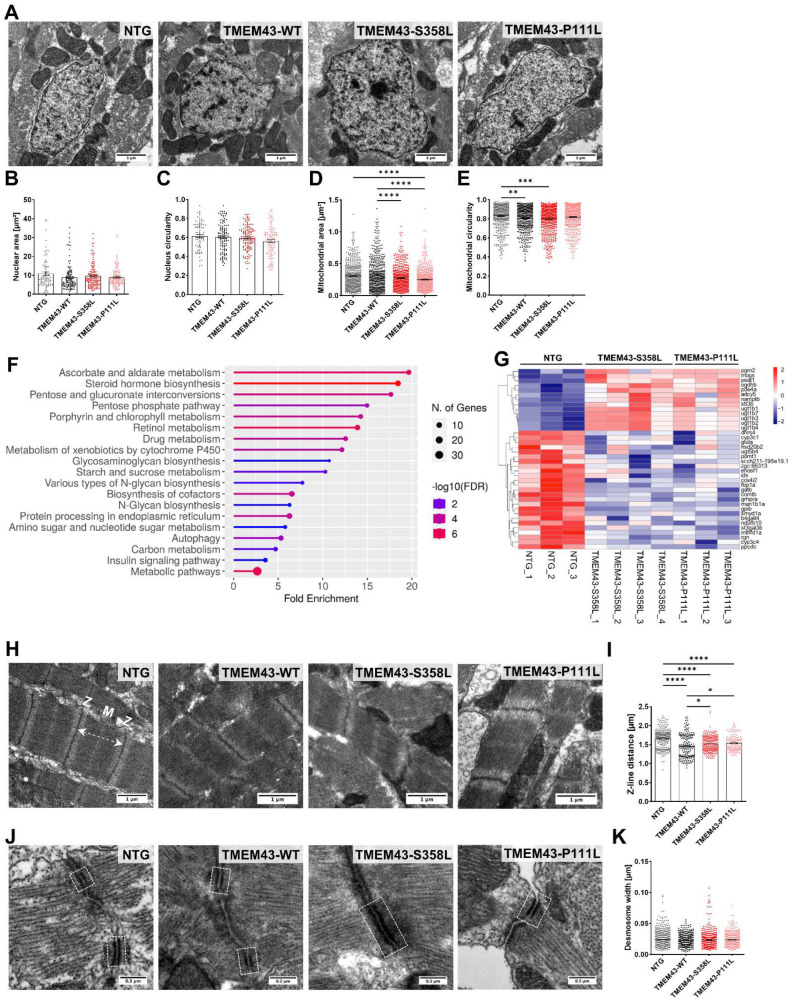
Transmission electron microscopy (TEM) reveals altered ultrastructure in ventricular heart tissue from TMEM43 transgenic lines at 5 months post-fertilization. (**A**) TEM images showing nuclei and mitochondria of the ventricular myocardium. Scale bars = 1 µm. (**B**) Analysis of the nuclear area and (**C**) circularity demonstrate no alterations between the genotypes. (**D**) Assessment of mitochondrial area uncovers significantly reduced mitochondrial size in both TMEM43 variant lines compared to TMEM43-WT. (**E**) Evaluation of mitochondrial circularity revealed altered mitochondrial shape in TMEM43-S358L and TMEM43-WT ventricles compared to the non-transgenic control (NTG). (**F**) Kyoto Encyclopedia of Genes and Genomes (KEGG) pathway enrichment analysis for shared differently expressed genes (DEGs) between “NTG vs. TMEM43-S358L” and “NTG vs. TMEM43-P111L” (p_adj._ ≤ 0.05) using ShinyGO v0.76. The x-axis represents fold enrichment and y-axis represents enriched KEGG terms. The size of the dots represents the number of involved genes. The false discovery rate (FDR) value is indicated by colors. (**G**) Heatmap showing DEGs belonging to the KEGG pathway “Metabolic pathways”. (**H**) Representative TEM images of longitudinal sections of sarcomeres showing proper myofibrill organization. Dashed double-headed arrow indicates distance between two Z-discs. M, M-line; Z, Z-line. Scale bars = 1 µm. (**I**) Quantification of Z-line distance of sarcomeres represented in (**H**) reveals a significantly increased sarcomere length in both TMEM43 variant lines compared to the TMEM43-WT. NTG displayed the highest sarcomere length. (**J**) TEM images showing desmosomes of the ventricular myocardium. Dashed boxes indicate desmosomal junctions. Scale bars = 0.3 µm. (**B**–**E**,**I**,**K**) For all graphs, significance was determined by one-way ANOVA with Bonferroni’s multiple comparison test, * *p* ≤ 0.05, ** *p* ≤ 0.01, *** *p* ≤ 0.001, **** *p* ≤ 0.0001. Error bars correspond to SEM. *n* ≥ 3 per genotype.

**Figure 7 ijms-23-09530-f007:**
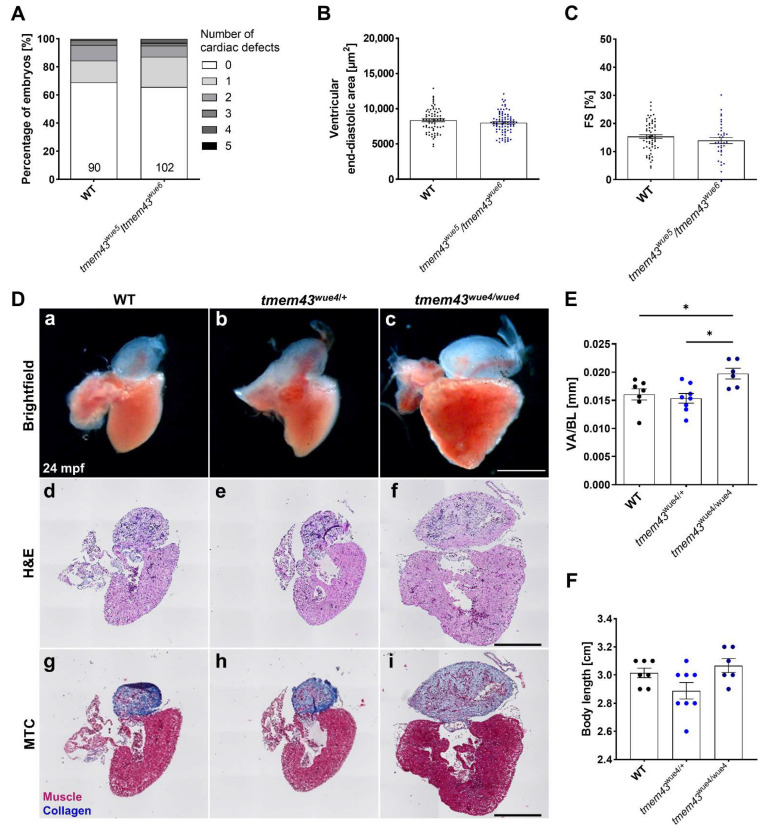
Enlarged adult hearts of homozygous tmem43^wue4/wue4^ mutant fish. (**A**) Phenotyping showing the percentage of embryos with zero to five different cardiac-specific defects. The defects were classified in looping defects, pericardial edemas, arrhythmias, altered ventricular shape, strong ventricular contractions, heterogenous ventricular contractions, and altered atrial shape. The number of analyzed embryos is indicated at the bottom of the columns. (**B**) Quantification of ventricular end-diastolic area showing the mean area of three individual sections of each fish from a time series of the embryonic cardiac cycle. (**C**) Analysis of cardiac function by quantification of ventricular fractional shortening (FS) from a time series of the embryonic cardiac cycle. Data points represent the average measurement obtained from three diastolic/systolic phases of each fish. The short axis of the ventricle was used for quantification. (**A**–**C**) Analyses were performed on embryos from three independent clutches per genotype. (**B**,**C**) Significance was determined by an unpaired t-test. Error bars correspond to SEM. (**D**) Adult cardiac morphology at 24 months post fertilization (mpf). Scale bars = 500 µm. (**a**–**c**) Representative brightfield images of dissected fish hearts. (**d**–**f**) Histology was analyzed by H&E staining (*n* = 5 hearts per genotype). (**g**–**i**) Fibrosis was analyzed by MTC staining (*n* = 5 hearts per genotype). (**E**) Quantification of ventricular area to body length ratio (VA/BL) reveals significant ventricular enlargement in homozygous tmem43^wue4/wue4^ zebrafish. Brightfield images represented in (**a**–**c**) were used for assessment of ventricular surface area. (**F**) Quantification of body length reveals normal growth and development in adult tmem43 mutants. (**E**,**F**) Significance was determined by one-way ANOVA with Bonferroni’s multiple comparison test, * *p* ≤ 0.05. Error bars correspond to SEM.

## Data Availability

The RNA-seq data are available at NCBI GEO (https://www.ncbi.nlm.nih.gov/geo (accessed on 18 July 2022)) under the accession number GSE208408.

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
