# Peer review of "Altered Expression of *TMEM43* Causes Abnormal Cardiac Structure and Function in Zebrafish"

_ijms, 2022, doi:10.3390/ijms23179530_

Round 1
Reviewer 1 Report
In this paper, authors investigated cardiomyocyte-restricted transgenic zebrafish lines that overexpress eGFP-linked full-length human wild-type (WT) TMEM43 and two genetic variants (c.1073C>T, p.S358L; c.332C>T, p.P111L) using the Tol2-system. Their aim is to study the role of TMEM43 in the pathogenesis of ACM.
The paper is very interesting and well written. I would like to suggest a brief comment in the Introduction on the challenges related to the differential dagnosis of ACM, see for example: "Antonini-Canterin F, Arrhythmogenic right ventricular cardiomyopathy or athlete's heart? Challenges in assessment of right heart morphology and function. Monaldi Arch Chest Dis. 2019 Mar 27;89(1). doi: 10.4081/monaldi.2019.1047. PMID: 30968653."
Author Response
Reviewer 1 comment:
The paper is very interesting and well written. I would like to suggest a brief comment in the Introduction on the challenges related to the differential diagnosis of ACM, see for example: "Antonini-Canterin F, Arrhythmogenic right ventricular cardiomyopathy or athlete's heart? Challenges in assessment of right heart morphology and function. Monaldi Arch Chest Dis. 2019 Mar 27;89(1). doi: 10.4081/monaldi.2019.1047. PMID: 30968653."
Answer:
Thank you for your constructive and helpful suggestion. We added the following comment in the Introduction (page 2, line 46):
As a consequence, clinical diagnosis of ACM remains challenging even with application of the task force criteria or Padua criteria. The diagnosis depends on a broad clinical assessment of patients with a combination of various types of information, including electro- and echocardiographic, histopathological and genetic data. Moreover, it is important to consider differential diagnosis that encompasses manifestations in an athlete's heart, cardiac syndromes or are common with other cardiomyopathies [1, 4, 6].
Reviewer 2 Report
Thank you for the opportunity to review your manuscript entitled "Altered Expression of TMEM43 Causes Abnormal Cardiac Structure and Function in Zebrafish".
Troponin T (TnT) is a protein forming part of the contractile apparatus of the striated muscle. The function of TnT in all types of striated muscles is the same, but cTnT is different from TnT found in skeletal muscles. Therefore, cTnT detected in plasma is a highly specific marker of myocardial damage (necrosis). High -sensitivity troponin tests, available for the past several years, detect troponin levels with a high degree of credibility. Numerous reports have demonstrated that the measurement of high-sensitivity TnT (hs -TnT) levels may have prognostic value in various cardiovascular disorders, such as sudden cardiac arrest (1).
Was the level of Troponin T determined in the study group?
REFERENCE
1. Doi: 10.20452/pamw.4107
Abstract, title and references.
The aim of the study is clear. The title is informative and relevant. The references are relevant, recent, and referenced correctly. Please add the following reference:
1. Doi: 10.20452/pamw.4107
Introduction.
It is clear what is already known about this topic. The research question is clearly outlined.
Methods.
The process of subject selection is clear. The variables are defined and measured appropriately. The study methods are valid and reliable. There is enough detail in order to replicate the study.
Discussion and Results
The results are discussed from multiple angles and placed into context without being overinterpreted. The conclusions answer the aims of the study. The conclusions supported by references and results. The limitations of the study are opportunities to inform future research.
Overall. The study design was appropriate to answer the aim. The manuscript is well written and a stimulus for the readership.
Minor revisions:
Was the level of Troponin T determined in the study group?
* Please add the following reference:
1. Doi: 10.20452/pamw.4107
Author Response
Reviewer 2 comment:
Troponin T (TnT) is a protein forming part of the contractile apparatus of the striated muscle. The function of TnT in all types of striated muscles is the same, but cTnT is different from TnT found in skeletal muscles. Therefore, cTnT detected in plasma is a highly specific marker of myocardial damage (necrosis). High -sensitivity troponin tests, available for the past several years, detect troponin levels with a high degree of credibility. Numerous reports have demonstrated that the measurement of high-sensitivity TnT (hs -TnT) levels may have prognostic value in various cardiovascular disorders, such as sudden cardiac arrest (1).Was the level of Troponin T determined in the study group? Please add the following reference: Doi: 10.20452/pamw.4107
Answer:
We agree with reviewer #2 that the detection of cardiac troponin T (cTnT) is a biochemical biomarker for myocardial injury for example after a myocardial infarct. However, we have not determined the level of cTnT in our study group. Due to the small size and blood volume of the zebrafish, it is difficult to collect enough blood to measure serum cTnT levels. To the best of our knowledge it is likewise unknown if the cTnT plasma level is increased in human patients carrying the TMEM43-p.S358L mutation. In contrast to myocardial injury caused by myocardial infarct, the development of ACM is a progressive but also slow pathological process leading to heart failure over several life decades in humans. Therefore, it is in our view unclear, if increased cTnT plasma levels could be really measured in our novel zebrafish model. In addition, cTnT meausrements do not provide any spatial information about the precise localization of the damaged heart. Therefore, we have investigated myocardial injury in the zebrafish heart by detailed histological and functional analyses (see Figure 5A and Figure S6).
Reviewer 3 Report
This manuscript describes a well designed experiment using transgenic zebrafish to determine how changes in the transcription of various genes affects cardiac development and function. The paper is well written. The comments below are suggestions regarding the addition of information that may help explain some of the results.
1) Do zebrafish every show fatty deposits in cardiac tissues? Some animals do not. Therefore the absence of fatty tissue in the heart may not be a marker for vascular disease or dysfunction in this model.
2) the authors suggest that genetic differences in amino acids at specific sites may underlie changes in cardiac muscle function. Have the authors examine how modulation of these sites, either by phosphorylation or acetylation might affect cardiac development and/or function?
Author Response
Reviewer 3 comments:
1) Do zebrafish every show fatty deposits in cardiac tissues? Some animals do not. Therefore the absence of fatty tissue in the heart may not be a marker for vascular disease or dysfunction in this model.
Answer: There is no evidence in the literature that zebrafish develop fatty deposits in the heart tissue. They have only been shown to develop epicardial fat, which is not associated with cardiac dysfunction (Brodehl A, Rezazadeh S, Williams T, Munsie NM, Liedtke D, Oh T, Ferrier R, Shen Y, Jones SJM, Stiegler AL, Boggon TJ, Duff HJ, Friedman JM, Gibson WT; FORGE Canada Consortium, Childs SJ, Gerull B. Mutations in ILK, encoding integrin-linked kinase, are associated with arrhythmogenic cardiomyopathy. Transl Res. 2019 Jun;208:15-29. doi: 10.1016/j.trsl.2019.02.004. Epub 2019 Feb 15. PMID: 30802431; PMCID: PMC7412573.). Of note, the most genetic ACM mouse models develop severe myocardial fibrosis – but without developing fatty replacement of the myocardial tissue (see for review Genetic Animal Models for Arrhythmogenic Cardiomyopathy. Gerull B, Brodehl A.Front Physiol. 2020 Jun 24;11:624. doi: 10.3389/fphys.2020.00624. eCollection 2020.PMID: 32670084). Nevertheless, since fibro-fatty remodelling is a hallmark of ACM in humans, we investigated this feature in our novel zebrafish model. However, we did not use the absence of fibro-adipose tissue as a general marker of vascular disease or dysfunction. We merely pointed out that this could be a possible reason for the lack of electrophysiological changes in the zebrafish.
2) The authors suggest that genetic differences in amino acids at specific sites may underlie changes in cardiac muscle function. Have the authors examine how modulation of these sites, either by phosphorylation or acetylation might affect cardiac development and/or function?
Answer: Unfortunately, we have not investigated how modulations of the different amino acid sites might affect cardiac development and/or function, so we can only speculate at this point. TMEM43-p.S358L affects a serine residue within a transmembrane domain. Therefore, it is highly unlikely, that a kinase is able to phosphorylate this serine residue. Therefore, we have not addressed this issue in our study.